# Coupling a single spin to the motion of a carbon nanotube

Federico Fedele [1,12], Federico Cerisola [1,2,12], Lea Bresque[3], Florian Vigneau[4], Juliette Monsel [5], Jorge Tabanera-Bravo[6], Kushagra Aggarwal [4], Jonathan Dexter[4], Sofia Sevitz[7], Joe Dunlop [2], Alexia Auffeves [8,9], Juan MR Parrondo [6], Andras Palyi[10,11], Janet Anders [2,7] & Natalia Ares [1] ✉

The ability to couple a solitary spin to high-frequency motion is a crucial advancement for a range of applications, including quantum sensing, intermediate and long-distance spin-spin coupling, and quantum information processing. Although proposed theoretically over a decade ago, experimental demonstrations have remained elusive. Here we report the observation of spin-mechanical coupling in a carbon nanotube device. We demonstrate this coupling in two configurations: off-resonant, with spin and mechanics excited by separate tones, and resonant, driven by a single tone. The coupling manifests as a shift and broadening of the electric dipole spin resonance (EDSR), respectively. Our theoretical model, which accounts for the tensor character of the coupling and mechanical non-linearity, reproduces the data with very good agreement. Our results demonstrate a previously unobserved spin-mechanical coupling, offering versatile tools for exploring macroscopic quantum phenomena, quantum thermodynamics, and quantum simulation.

Coupling a single spin to the motion of a carbon nanotube (CNT) presents a promising avenue for applications in quantum technologies[1–7]. Due to their small mass and unique material properties, CNTs can realise resonators with quality factors of over a million[8] and resonant frequencies of up to 39 GHz[9]. With such high resonance frequencies, CNT motion can reach the quantum regime at dilution refrigerator temperatures[10,11]. CNTs can also extend over macroscopic distances while still exhibiting extraordinary mechanical properties. This means that realising coupling between spin and motion will be particularly useful to connect well-separated locations on chip and to store quantum information, enabling hybrid quantum networks[12]. Unlike most of the alternatives for intermediate and long-range spin

coupling, mechanical resonators can exhibit longitudinal parametric coupling to spins[13], enabling low power entangling operations[14] and fast quantum-non-demolition readout of the spin states[15]. From a quantum foundations perspective, spin-mechanical coupling is highly desired to enable teleportation and entanglement swapping of macroscopic states of motion[16,17]. This platform is also promising for the exploration of the quantum-to-classical transition[18].

The interaction between single spins and mechanics has previously been demonstrated for a single NV spin-qubit probed by a magnetized AFM cantilever[4,19], as well as in monolithic platforms, where single NV spin qubits are embedded in cantilevers and semiconductor nanowires[13,18,20–22], and with an InGaAs quantum dot[23]

[1]Dept. of Engineering Science, University of Oxford, Oxford, UK. [2]Dept. of Physics and Astronomy, University of Exeter, Exeter, UK. [3]Univ. Grenoble Alpes, CNRS, Grenoble INP, Institut Néel, Grenoble, France. [4]Dept. of Materials, University of Oxford, Oxford, UK. [5]Dept. of Microtechnology and Nanoscience (MC2), Chalmers University of Technology, Göteborg, Sweden. [6]Dept. of Structure of Matter, Thermal Physics and Electronics and GISC, Universidad Complutense de Madrid, Madrid, Spain. [7]Institute of Physics and Astronomy, University of Potsdam, Potsdam, Germany. [8]MajuLab, CNRS-UCA-SU-NUS-NTU International Joint Research Laboratory, Singapore, Singapore. [9]Centre for Quantum Technologies, National University of Singapore, 117543 Singapore, Singapore. [10]Dept. of Theoretical Physics, Institute of Physics, Budapest University of Technology and Economics, Muegyetem rkp. 3, H-1111 Budapest, Hungary. [11]HUN-REN-BME-BCE Quantum Technology Research Group, Budapest University of Technology and Economics, Budapest, Hungary. [12]These authors contributed equally: Federico Fedele, Federico Cerisola. ✉e-mail: natalia.ares@eng.ox.ac.uk

hosted in a GaAs cantilever. These mechanical oscillators have resonant frequencies below 6 MHz and thus high phonon occupancies even at cryogenic temperatures. Cooling mechanical motion can be achieved using a variety of techniques[24], including laser cooling[25]. However, these techniques are often incompatible with qubit operation, for example, due to substrate heating[26]. Extensive theoretical and experimental research has been dedicated to finding platforms in which single spins could couple to mechanical resonators that can be operated at high frequencies and thus access the quantum regime[27–29].

CNT devices arise as a particularly promising platform. Electrons/holes can be confined in quantum dots, which are defined electrostatically in the CNT[30–32]. The coupling between single-electron tunneling and the CNT motion was recently found to reach the ultrastrong coupling regime[33,34]. However, while coupling the spin degree of freedom to the mechanical motion has been predicted theoretically[35], it has never been demonstrated. Here, we report on the first ever observation of such coupling. The single spin is driven by electrically driven spin resonance (EDSR) at radio-frequencies and interacts with the nanotube's motion via spin-orbit coupling. We observe this effect when the Larmor and mechanical frequencies are both, resonant and off-resonant. We develop an advanced theoretical model that captures the dependence of the gyromagnetic tensor on the CNT displacement, including non-linearities in the nanotube's motion. Our side-by-side figures show extraordinarily good agreement between experimental observations and theory predictions for this nanoscale system.

## Results

The CNT is stamped across metallic contact electrodes to give a vibrating segment of length ~ 900 nm, and is measured at a

temperature of 45 mK. Voltages applied to five finger gates beneath the nanotube (labeled $V_{G1}$–$V_{G5}$) are used to form a double quantum dot (DQD) electrostatically (Fig. 1A). A voltage bias $V_{SD}$ is applied between the leads to drive a DC current $I$. Both single and DQD configurations are accessible (see Section S1 of the Supplementary Information). In the DQD regime, we focus on a charge transition that exhibits Pauli spin blockade. With an external magnetic field $B_X$, Pauli spin blockade is observed as an enhancement of the triangle baseline due to selection rules on spin and valley states (Fig. 1B, C)[36]. Pauli spin blockade is identified by a suppression instead of an enhancement of the triangle baseline, with the latter signature found only in systems with strong-spin orbit coupling[37–40]. A cycle of gate voltage pulses applied to G3 and G5, see Fig. 1D, is used to define and control a two-level system, identified as a spin-valley qubit, using EDSR[41]. First, G3 and G5 are pulsed to the gate voltage coordinate indicated by a square marker in Fig. 1C for 250 ns, where an effective triplet state is initialised by configuring the double dot in Pauli spin blockade. Then, G3 and G5 are pulsed back to the gate voltage coordinate marked with a triangle to set the double dot in Coulomb blockade for 500 ns, while a microwave burst with a duration of 400 ns is applied to either G2 or G4 to manipulate the spin-valley state. Finally, G3 and G5 are pulsed back to the Pauli spin blockade configuration for another 250 ns. The total pulse cycle duration was 1 μs. If the spin-valley state was flipped during the microwave burst, Pauli spin blockade is lifted and the current changes. The resulting change in current $\Delta I$ as a function of the frequency of the microwave drive burst $f_{d,spin}$ and magnetic field $B_X$ is shown in Fig. 1E. Dips in $\Delta I$ appear as diagonal lines indicating a resonance when $f_{d,spin}$ matches the qubit frequency $f_{EDSR}$. From the slopes of these diagonal lines, we extract effective g-factors: $g_X^{(1)} =$

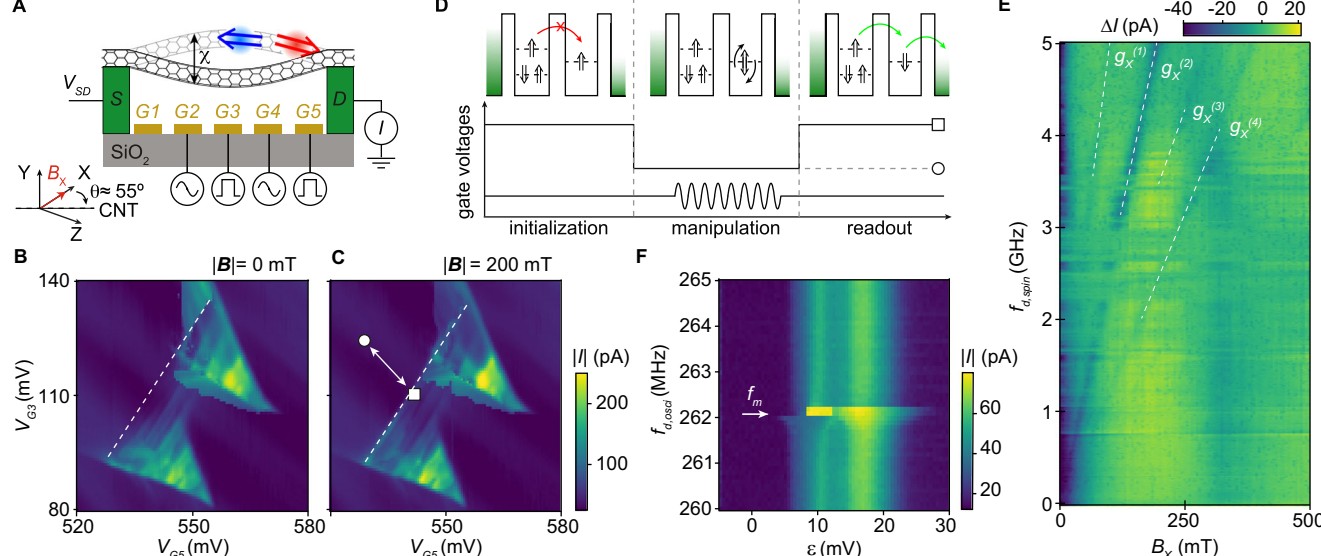

**Fig. 1 | Determination of spin-valley and mechanical frequencies. A** The carbon nanotube is suspended between source and drain contacts, and over five gate electrodes, G1–G5. We measure a current $I$ through the device against a source-drain voltage $V_{SD}$. A magnetic field $B_X$ is applied along the X direction, with the CNT oriented at an angle $\theta \approx 55°$ relative to the X axis (see Section S2 of the Supplementary Information). A double quantum dot confined along the CNT (red and blue double-arrows) allows us to define a qubit and enables readout via Pauli spin blockade. The CNT motion is driven by a microwave drive at frequency $f_{d,osci}$, resulting in an amplitude $\chi$. **B, C** Absolute value of the current $|I|$, measured as a function of gate voltages $V_{G3}$ and $V_{G5}$ at a Pauli-blocked transition, with magnetic field strength $|\mathbf{B}| = 0$ mT and $|\mathbf{B}| = 200$ mT, respectively. A white dashed line marks the baseline of the bias triangles. The square (circle) indicates the initialisation/readout (manipulation) gate voltages, and the arrow marks the pulse axis, see schematic D. **D** Pulse scheme. Upper panel: Double-dot occupations illustrating

how Pauli spin blockade makes the current sensitive to the spin-valley state (double-arrows). Lower panel: Gate-voltage pulse cycle for qubit initialisation, manipulation, and EDSR detection. A microwave drive at frequency $f_{d,spin}$ is applied during the manipulation stage. **E** $\Delta I$ as a function of drive frequency $f_{d,spin}$ and magnetic field $B_X$. A microwave drive of power $P_{d,spin} = -33$ dBm at the device is applied to gate G4. The average current in each row is subtracted to highlight the EDSR resonances ($f_{EDSR}$). Resonances appear as dark lines (dashed white lines are a guide to the eye). **F** Observation of the nanotube's mechanical frequency ($f_m$), identified at ca. 261.9 MHz, by measuring the current $I$ as a function of double dot detuning ($\varepsilon$) while sweeping $f_{d,osci}$. The drive power $P_{d,osci}$ (applied to G4) is −33 dBm at the device and the field is $B_X = 200$ mT. The detuning axis and reference point (defined from the triangle baseline) differ from those in (**C**). No EDSR excitation was applied for this measurement.

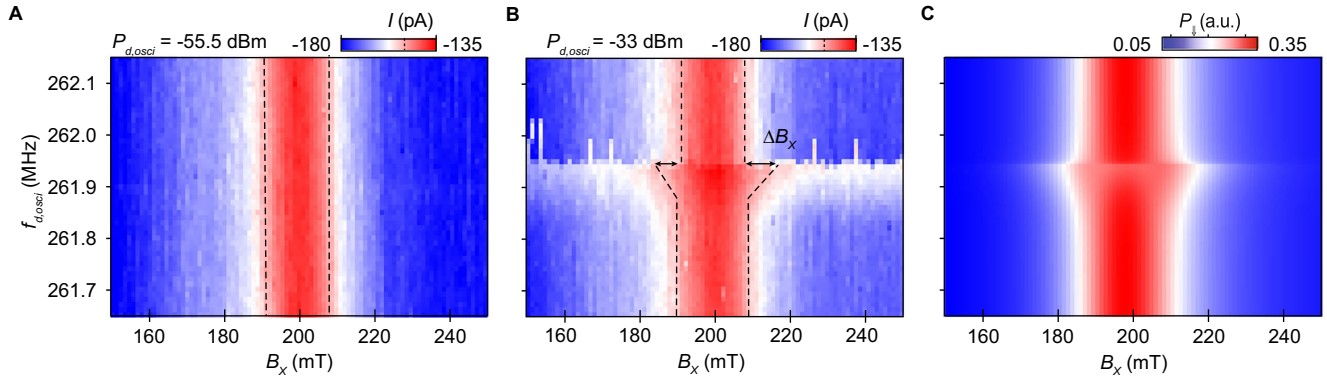

**Fig. 2 | Off-resonant coupling between a spin-valley qubit and mechanical motion. A, B** Measured EDSR resonance at a field $B_X \approx 200$ mT corresponding to a drive at $f_{d,spin} \sim 5.1$ GHz, while the carbon nanotube motion is excited (at G2) at frequencies $f_{d,osci}$ in the range [261.7, 262.1] MHz. The resonance is the one with g-factor $g_X^{(2)}$ in Fig. 1. Here, the spin manipulation is performed using a continuous microwave drive (at G4) with power $P_{d,spin} = -34.5$ dBm. **A, B** correspond to different mechanical drive powers $P_{d,osci}$, $-55.5$ dBm and $-33$ dBm, respectively. For the larger value of $P_{d,osci}$ (**B**), we observe EDSR resonance broadening ($\Delta B_X^{EDSR}$ indicated by arrows) at $f_m \sim 261.9$ MHz, the frequency corresponding to the mechanical resonance. To quantify this broadening, we pinpoint the values of $B_X$ corresponding to a reduction of $1/\sqrt{2}$ of EDSR current value (dashed black lines). **C** shows the calculated transition probability $P_\Downarrow$ from the blocked to the unblocked state, time-averaged until the steady state is reached. The transition probability is obtained by solving the dynamics of Hamiltonian (2) up to first order in $\chi$. Simulation parameters are provided in Section S7 of the Supplementary Information. Our simulation reproduces the EDSR broadening observed in (**B**).

$(3.30 \pm 0.30)$, $g_X^{(2)} = (1.84 \pm 0.02)$, $g_X^{(3)} = (1.31 \pm 0.06)$ and $g_X^{(4)} = (0.94 \pm 0.01)$. Errors reflect the uncertainty of a linear fit to the EDSR resonances. These multiple EDSR resonances $g_X^{(j)}$ can be attributed to transitions within different subsets of spin-valley states ($g_X^{(1)}$ and $g_X^{(3)}$), as well as higher harmonics ($g_X^{(2)}$ and $g_X^{(4)}$). We actuate vibrations by injecting a radio-frequency tone with driving power $P_{d,osci}$ to either G2 or G4. An abrupt change in $I$ is observed when the frequency of the mechanical drive tone $f_{d,osci}$ is close to the resonance frequency of the mechanical resonator at ca. $f_m = 261.9$ MHz (Fig. 1F).

## Off-resonant spin-mechanical coupling

In order to evidence the spin-mechanical coupling, we monitor the EDSR resonance while actuating the nanotube's motion (Fig. 2A, B). Specifically, we fixed $f_{d,spin}$ at 5.1 GHz, thus driving the EDSR resonance corresponding to $g_X^{(2)}$ in the vicinity of $B_X^{EDSR} = 200$ mT, and swept $f_{d,osci}$ in a range of frequencies around $f_m$. For large enough values of $P_{d,osci}$, we observe a broadening $\Delta B_X$ of the EDSR resonance at $f_{d,osci} \sim f_m$. To quantify this broadening, we identify the values of $B_X$ for which the current drops from the maximum EDSR current value by a factor of $1/\sqrt{2}$. The dependence of the broadening $\Delta B_X$ on the power $P_{d,osci}$ is discussed further in section S3 of the Supplementary Information. We observe that the broadening $\Delta B_X$ starts at slightly lower frequencies and hits the maximum at $f_m$. In contrast, for frequencies slightly larger than $f_m$, no broadening is observed. This asymmetry becomes more evident as $P_{d,osci}$ increases, indicating the presence of non-linearities in the mechanical motion, which are expected to become more pronounced at strong driving powers[42–44]. For instance, mechanical resonance hardening, i.e., an increase of $f_m$, and the occurrence of current switches that might be revealing mechanical bistabilities, are further evidence that mechanical non-linearities are present (see Section S3 of the Supplementary Information). It is important to note that for Fig. 2A, B, $f_{d,osci}$ is swept from high to low frequencies.

## Theoretical model

We model the impact of the interaction between the mechanics and the spin-valley qubit as a modulation of the qubit frequency $f_{EDSR}$. The qubit is subjected to the applied magnetic field $B_X$ with components parallel ($B_\parallel$) and perpendicular ($B_\perp$) to the CNT's axis at zero displacement. The presence of spin-orbit coupling may induce an anisotropic gyromagnetic tensor[35,45,46], $\mathbf{g}^{(j)}$, with components $g_\parallel^{(j)}$ and $g_\perp^{(j)}$, where the superscript j indicates the different spin-valley resonances

observed in Fig. 1E and their associated resonances. For the CNT at rest, the qubit Hamiltonian for a given pair of spin-valley states is

$$H_{rest} = \frac{1}{2}\mu_B \sqrt{B_\parallel^2 g_\parallel^2 + B_\perp^2 g_\perp^2}\ \sigma_3, \tag{1}$$

where $\sigma_3$ is the Pauli operator in the energy quantisation axis of the qubit. When the mechanical motion is driven, the CNT displaces with an amplitude $\chi$ and $B_\parallel$ and $B_\perp$ in Eq. (1) can be modified as follows: $B_\parallel \rightarrow (B_\parallel \ell - B_\perp \chi)/\sqrt{\chi^2 + \ell^2}$, and $B_\perp \rightarrow (B_\perp \ell + B_\parallel \chi)/\sqrt{\chi^2 + \ell^2}$, where $\ell$ is the distance between the qubit and the closest lead. For small oscillation amplitudes $\chi$, we then expand to second order and obtain the qubit Hamiltonian

$$H = H_{rest} + \lambda_1 \chi\, \sigma_3 + \lambda_2 \chi^2\, \sigma_3, \tag{2}$$

where $\lambda_1$ and $\lambda_2$ are corrections to the qubit frequency that depend on $g_\parallel$, $g_\perp$, and the field orientation (see Section S2 of the Supplementary Information).

Using Eq. (2), we can calculate the steady state qubit transition probability $P_\Downarrow$ to the unblocked state that is proportional to the current flowing through the CNT, $I$. Although not at the heart of the spin-mechanical coupling, the signatures of mechanical non-linearities observed in Fig. 2A, B, motivate the use of a Duffing model for the mechanics, see section S3 of the Supplementary Information. Fig. 2C shows $P_\Downarrow$ estimated from the Hamiltonian corresponding to Eq. (2) up to first order. For this numerical simulation, we set the linear coupling parameter $\lambda_1$ to 433 neV/nm, which we later find to be consistent with the range of values extracted from fitting the measurement shown in Fig. 3A. Additional simulation parameters are provided in Section S7 of the Supplementary Information. Including just the first order correction in $\chi$ the model captures the physics of the off-resonant experiment, see Fig. 2B. We thus conclude that the main effect of the off-resonant coupling between the spin-valley qubit and the mechanical motion is a broadening of the EDSR resonance. The broadening of $f_{EDSR}$ is governed by the product of the coupling strength and the mechanical amplitude $\lambda_1 \chi / h$, where $h$ is the Planck constant.

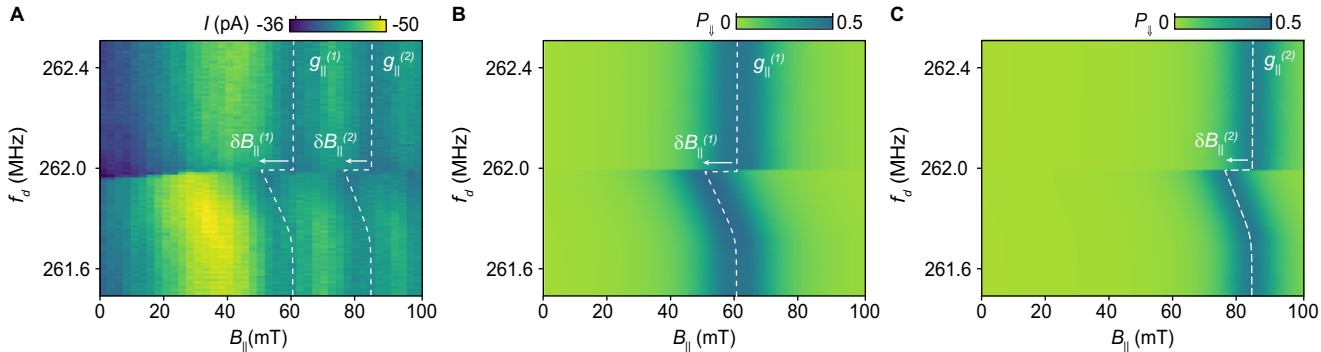

**Fig. 3 | Resonant coupling between a single spin and mechanical motion.**
**A** Measured resonant signal $I$ as a function of the applied parallel field $B_\parallel$ and a single drive frequency $f_d$ at power $P_d = -33$ dBm. The white dashed lines are a guide to the eye for the EDSR resonances. For values of $f_d$ far from $f_m$, these resonances corresponds to g-factors of $g_\parallel^{(1)}$ and $g_\parallel^{(2)}$. At $f_d$ slightly below $f_m = 261.9$ MHz, the EDSR resonances shift significantly towards lower values of $B_\parallel$. White arrows indicate these shifts, $\delta B_\parallel^{(1)}$ and $\delta B_\parallel^{(2)}$. Here, $I$ is detected by chopping the drive signal applied via $G2$ at 83.1 Hz and locking into the chopper signal[62]. **B**, **C** Calculated transition probability $P_\downarrow$ from the blocked to the unblocked state, time-averaged until the steady state is reached. The transition probability is obtained by solving the dynamics of Hamiltonian (2) up to second order in $\chi$ plus the EDSR driving tone, and including decoherence effects due to the environment (see Section S7 of the Supplementary Information for details). Our simulation for the $g_\parallel^{(1)}$ and $g_\parallel^{(2)}$ resonances reproduce the EDSR resonances observed in (**A**).

## Resonant spin-mechanical coupling

So far we have demonstrated spin-mechanical coupling by measuring the broadening $\Delta B_X^{EDSR}$ of the EDSR peak at $f_m$, see Fig. 2. This was achieved in an off-resonant regime, $f_{EDSR} > f_m$. The model developed to account for spin-mechanical coupling, see Eq. (2), considers a modulation of the Zeeman Hamiltonian that depends on the magnetic field orientation and the g-factor anisotropy. To further test the nature of this coupling, we now align the magnetic field with the CNT axis (see Section S2 of the Supplementary Information). Our model in Eq. (2) predicts that the first order correction $\propto \lambda_1$ is switched off and the Zeeman energy modulation is now dominated by the term quadratic in $\chi$ (see Section S7 of the Supplementary Information). Moreover, for this magnetic field orientation, we are able to access the resonant spin-mechanical coupling regime, i.e., $f_{EDSR} \sim f_m$. This is because in this orientation, the g-factor's value is reduced, thus shifting the EDSR resonance condition for $f_m$ to higher fields where Pauli spin blockade is not lifted. At the resonance condition, we can drive both the EDSR resonances and the mechanical motion with a single drive tone $f_d$ applied to G2.

In Fig. 3A we display the EDSR resonances corresponding to g-factors $g_\parallel^{(1)} = 0.31 \pm 0.03$ and $g_\parallel^{(2)} = 0.22 \pm 0.02$ (these are the g-tensor components aligned with the CNT axis, while those in Fig. 1E are the X-aligned components). For $f_d \sim f_m$, we observe a shift of the EDSR resonances, $\delta B_\parallel^{(j)} = B_\parallel^{(j)}(f_m + \epsilon) - B_\parallel^j(f_m - \epsilon)$ for small $\epsilon > 0$ with $j = 1, 2$. These shifts show a strong asymmetry: at frequencies $f_d$ smaller than $f_m$, the EDSR resonance is shifted to lower field strengths, reaching the maximum shift at $f_m$. In contrast, for frequencies only slightly larger than $f_m$, no shift is observed. (Note that for Fig. 3A, $f_d$ is swept from high to low frequencies.)

To model the observed frequency shift, we now require in Eq. (2) the second-order correction that arises from the spin-mechanical coupling. For each EDSR resonance, this is given by

$$\lambda_2 = E_{rest} \frac{g_\perp^2 - g_\parallel^2}{4\ell^2 g_\parallel^2}, \tag{3}$$

where $E_{rest}$ is the energy gap of $H_{rest}$, see Eq. (1). We note that this second-order term arises from a longitudinal-type coupling, in contrast to the transverse interaction characteristic of the Jaynes-Cummings model (see Section S7 of the Supplementary Information for further details). The time average effect of this coupling is to produce an effective qubit frequency given by

$$f_{eff} = \frac{\mu_B}{h} g_\parallel B_\parallel + \frac{1}{2h} \lambda_2 A^2(f_{G2}), \tag{4}$$

where $A(f_{G2})$ is the frequency response of the mechanical oscillator considering the Duffing non-linearity. This expression allows us to extract the value $\lambda_2 A_{max}^2$, with $A_{max}$ the maximum displacement, from the fit of the EDSR resonances from Fig. 3A (white dashed line). Fig. 3B shows the simulation of the transition probability using the parameters obtained from the fit of Eq. (4). We find that the experimental measurements, Fig. 3A, are extremely well matched with the theoretical model, Fig. 3B, C. This agreement unequivocally proves that the observed frequency shift in the resonance is due to spin-mechanical coupling.

The value of $\lambda_2$ estimated from the fit, together with Eq. (3) allows us to make a quantitative prediction of the $g_\perp$ coefficient. Here one needs to include an estimate of the qubit's position $\ell$ on the CNT, which is ca. 900 nm long. For a reasonable range of $\ell$, i.e., $\ell = 50-250$ nm, we find $g_\perp = 4-24$. Note that the coupling corrections $\lambda_1$ and $\lambda_2$ are field orientation dependent. However, having found the two g-tensor values, $g_\parallel$ and $g_\perp$, we can now determine these corrections for arbitrary field orientations. In particular, for the magnetic field applied along the X direction, and accounting for the uncertainty in the mechanical displacement of the CNT, we find $\lambda_1 = 380-455$ neV/nm. This result is consistent with the value of 433 neV/nm used in the simulation of Fig. 2C.

## Discussion

Here we have demonstrated spin-orbit-mediated coupling of a spin-valley qubit to high-frequency motion in a suspended CNT device[47]. As seen in Figs. 2 and 3, coupling is possible both when the qubit and mechanics are off-resonant, as well as when they resonate at the same frequency. The good agreement between theory and experiment allows us to fully map out the coupling mechanism at play. By probing this coupling for two different orientations of the applied magnetic field, we uncover that the coupling strength is strongly affected by the g-factor anisotropy, indicating that our approach provides a promising tool to investigate such anisotropy in CNTs[48-51]. We also give an experimental estimate of the strength $\lambda_1$ of the spin-mechanical coupling.

The spin-mechanical coupling we demonstrate unlocks experiments combining single spin qubits with a linear or non-linear, classical or quantum resonator. A single spin strongly coupled to a quantum resonator acting as a battery allows, for example, the exploration of quantum batteries[52–54], quantum Maxwell demons[55] and thermodynamic engines[56]. For highly coherent spin states[57,58], spin-mechanical coupling would allow for work extraction from quantum coherence[59] and the realisation of macroscopic states of motion[17]. It could also enable ground-state cooling of a macroscopic resonator by spin-polarized currents[60]. The coupling of high-frequency mechanics and spin offers novel approaches for high-efficiency microwave-to-optical conversion[61] and mid- to long-range coupling of spin qubits[3,6].

## Methods

### Device fabrication

The CNT used in this experiment has been grown by chemical vapor deposition and then transferred via flip-chip stamping on a silicon chip. For the CNT synthesis, we deposit a $FeCl_3$ catalyst mixed with PMMA on a quartz substrate with etched pillars. The CNT are then grown in a furnace with a $CH_4/H_2$ atmosphere diluted to 20% concentration in Argon at 950 °C. After the growth, the CNTs are transferred to the device chip by stamping using an optical mask aligner. The device layout consisting of two 110 nm tall Cr/Au pillars and five 18 nm tall Cr/Au gate electrodes was realised on the silicon substrate using standard electron beam lithography followed by thermal evaporation of Cr/Au gates. All five gates underneath the CNT are connected to bias tees that allow the application of DC and high bandwidth AC voltages. The length and the radius of the CNT used for this experiment have not been measured directly; however, these parameters have been estimated from previous experiments and found to be, respectively, $936 \pm 10$ nm and $3.9 \pm 0.2$ nm[33]. Specifically, the mass, radius and length of the CNT have been obtained by fitting the dependence of the mechanical resonance from the gate voltage. The device was measured inside a Triton 200 cryofree dilution refrigerator with a base temperature of 45 mK, and equipped with a vector magnet that could generate 6 T along the cryostat main axis, defined as Y in the laboratory frame of coordinates, and 1 T along the remaining axes X and Z.

## Data availability

The datasets generated and analysed during the present work are available at [ref. 47].

## Code availability

The codes used for the analysis of the present work are available at [ref. 47].

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

## Acknowledgements

We thank D. Zumbhül, E. Laird and G. J. Milburn for stimulating discussions on the subject of this research. This research was supported by grant number FQXi-IAF19-01 from the Foundational Questions Institute Fund, a donor-advised fund of the Silicon Valley Community Foundation. NA acknowledges the support from the Royal Society (URF-R1-191150), EPSRC Platform Grant (grant number EP/R029229/1) and the European Research Council (ERC) under the European Union's Horizon 2020 research and innovation programme (grant agreement number 948932). A.P. acknowledges support from the HUN-REN Hungarian Research Network, by the Ministry of Culture and Innovation and the National Research, Development and Innovation Office (NKFIH) within the Quantum Information National Laboratory of Hungary (Grant No. 2022-2.1.1-NL-2022-00004), and by NKFIH via the OTKA Grant No. 132146. J.M. acknowledges funding from the Swedish Research Council (VR) (Project No. 2018-05061). S.S. and J.A. acknowledge support by the DFG Research Unit FOR 2724 on "Thermal machines in the quantum world". J.A. is supported by EPSRC (EP/R045577/1), and thanks the Royal Society for support.

## Author contributions

F.F. led the experiment, with contributions from F.V. and N.A. F.V. fabricated the device. F.C. and A.P. led the development of the theoretical model, with contributions from F.F. L.B., J.A. and N.A. F.C. and F.F. led the data analysis with contributions from L.B., K.A. and S.S. J.M., J.TB., J.Dexter, J.Dunlop, A.A. and J.MR.P. contributed to the discussion, interpretation and presentation of the results. F.F., F.C., J.A. and N.A. wrote the manuscript with inputs from all the authors. N.A. designed and supervised the experiment.

## Competing interests

The authors declare no competing interests.
