## [Transparent Peer Review file · Nature Communications]

Coupling a single spin to the motion of a carbon nanotube

Corresponding Author: Professor Natalia Ares

Version 0:

Reviewer comments:

Reviewer #1

(Remarks to the Author)

The authors report on the coupling between spin and mechanics in a suspended carbon nanotube. Carbon nanotube mechanical resonators are interesting since their small mass enables large zero point motion, which boosts the coupling between the mechanical vibrations and other degrees of freedom. The mechanical frequencies can be particularly large so that they reach the mechanical quantum ground state in a dilution fridge. On the other hand, the spin physics in nanotubes is rich and has attracted a lot of attention (see for instance, Rev. Mod. Phys., Vol. 87, No. 3, July–September 2015). Here, the authors report the first measurements indicating the coupling of vibrations and spin in a nanotube. Merging the physics of mechanics and spin in nanotubes (and graphene) is important and opens possibilities for new quantum experiments.

The authors show data indicating the presence of Pauli-spin blockade and the formation of a spin-valley qubit. By driving the system at RF frequencies through the gates, they see features in the transport current at the mechanical frequency of the carbon nanotube and compare these features with a theoretical model where the non-linear mechanical classical motion is coupled to the qubit. The results are interesting and the theoretical model matches well the data they present.

The manuscript is well written overall. This work may perhaps be published in a high-impact journal such a Nature Communications provided that the authors respond the following questions in a convincing way.

- 1- While the data in Fig.2B can be related to spin-mechanics coupling, is it possible to explain the data by charge-mechanics coupling without any direct spin-mechanics coupling? The mechanical vibrations are observed away from the EDSR resonance, indicating charge-mechanics coupling. The combination of the electrical current shift due to mechanical vibrations (away from the EDSR resonance frequency) and of the current related to the EDSR resonance (away from the mechanical frequency) could lead to the impression of some sort of coupling even if there is not spin-mechanics coupling.
- 2- Is Fig.3a which is associated by the authors to a resonant coupling between the mechanical vibrations and the spin (as claimed in the Fig3 title), are the data really indicating resonant coupling in the sense that e.g. the quantum is oscillating between the mechanics and the spin? Or are the measurements related to the interplay between the drive of the mechanics and the drive of the spin without any direct spin-mechanics coupling? These are fundamentally two different phenomena.
- 3- The coupling λ_1 is given with high precision 433 neV/nm. What is the error bar? How is calibrated the mechanical displacement?
- 4- Why is it necessary that the nanotube is brought to the non-linear regime (Duffing) to see a signature of the spin-mechanics coupling?

These questions that are less critical should also be addressed.

- 5- What are the free parameters in the theoretical model to fit the power-dependent data in Fig. 2?
- 6- The text reads "the charge transition $(N,1)-(N+1,0)$ " in a p-n configuration. We understand that $(N,1)$ corresponds to N holes in the left dot and 1 electron in the right dot. Should the text not read $(N,1)-(N-1,0)$?
- 7- In Fig 1F, how do you define experimentally zero detuning?
- 8- (line 191): How can the nanotube length and radius be estimated from previous experiments?
- 9- (line 29) At least one reference should be provided
- 10- (line 58) Looking at the charge stability diagram of extended data Fig.1, it is not clear that the transition is $(N-1,1) \rightarrow (N,0)$
- 11- (Line 61) What is the orientation of the magnetic field B_X ? Is it along the nanotube axis? Has this been calibrated? Have the authors tried some other orientations? There is some extra information in the supplemental material but this should be

clearer in the main text.

12- (Line 62) The suppression of the triangle baseline is not at all clearly seen in figure 1B/C.

13- (Line 66) In order to state a spin-valley qubit is being defined, either spectroscopy or time-resolved measurements of the qubit should be provided, also giving information about its corresponding decoherence and possible multilevel character.

14- (Line 66) I recommend adding some more details clarifying the time-resolved protocol.

15- (Line 80) In Figure 1F, is the feature observed at 262 MHz the only mechanical resonance in the entire frequency spectrum?

16- (Line 116) The authors should stay at some point that the mechanical non-linearity is probably due to the high driving power and not to the coupling of the spin.

Reviewer #2

(Remarks to the Author)

Reviewer #3

(Remarks to the Author)

In this manuscript the authors demonstrate the coupling of a single spin with the bending motion of a carbon nanotube. Several spin transitions are identified via EDSR. The coupling between such spin transitions and the mechanical degree of freedom of the nanotube is observed in two distinct regimes. In the off-resonant regime, the considered spin transition is a few GHz away from the mechanical bending mode, and the main observed effect is a mechanically induced broadening of the EDSR resonance. In the resonant regime, two neighbor spin transitions could be brought into resonance with the mechanical mode and both of them undergo a frequency shift of their associated EDSR peak. The authors also provide a model describing the spin-mechanical coupling relying on the anisotropy of the g-factor. This model could explain very well both observed behaviors; namely, an increased dissipation in the non-resonant regime and a frequency shift in the resonant regime.

I think those results are remarkable and represent a long-standing goal in the community. I strongly support the publication of this work in a high impact journal such as Nature Communication provided the authors answer the following questions:

- The mechanical origin of the signal at around 262MHz is attested by the bifurcation behavior which is characteristic of bending mode in carbon nanotubes. Do you have extra data about the mechanical resonance, such as power dependence or a broad range frequency scan? Such extra measurements could be useful to know whether it is the fundamental mode or not.
- Could you give an estimation of the spin-orbit coupling in the nanotube based on the g-factor anisotropy? Since this anisotropy is at the core of the coupling mechanism, it could be interesting to discuss its origin a bit more.
- From the charge stability diagram presented in the extended data figure 1, it is not clear to me where is the semiconducting gap along the gate Vg5. Estimating an absolute electron number seems a bit audacious here. A broader map could maybe help to identify with more reliability the p-n and p-p quadrant. Do you have such a 2D-map?
- In figures 2A, 2B, and 3A the frequency is swept from high to low. Why? I would expect a stronger signal if it were swept in the opposite direction due to the Duffing behavior of the bending mode of the nanotube.
- I found the naming of some frequencies a bit confusing, such as f_{osci} and f_{spin} which are two driving frequencies. A subtext "d,..." or "drive,..." would clarify their meaning. Same for P_{osci} .

Version 1:

Reviewer comments:

Reviewer #1

(Remarks to the Author)

The authors investigate spin-mechanics coupling in suspended carbon nanotubes. Owing to their low mass and large zero-point motion, nanotube resonators enable strong coupling to other degrees of freedom, while their high mechanical frequencies allow the possibility of ground-state operation at dilution temperatures. Nanotube spin physics is also rich and well established. This work presents the first experimental evidence of spin-vibration coupling. The authors observe Pauli-spin blockade and spin-valley qubit formation. The authors present two sets of data, one called off-resonant coupling and the other called resonant coupling, and these two data sets can be described by a model using the same parameters. The manuscript is clearly written and I recommend publication. However, the outlook in the abstract and conclusion seems somewhat overstated in light of the present results.

Reviewer #2

(Remarks to the Author)

Reviewer #3

(Remarks to the Author)

I am satisfied with the reply of the authors and fully recommend the publication in Nature Communications.

Reviewer 1

The authors report on the coupling between spin and mechanics in a suspended carbon nanotube. Carbon nanotube mechanical resonators are interesting since their small mass enables large zero point motion, which boosts the coupling between the mechanical vibrations and other degrees of freedom. The mechanical frequencies can be particularly large so that they reach the mechanical quantum ground state in a dilution fridge. On the other hand, the spin physics in nanotubes is rich and has attracted a lot of attention (see for instance, Rev. Mod. Phys., Vol. 87, No. 3, July–September 2015). Here, the authors report the first measurements indicating the coupling of vibrations and spin in a nanotube. Merging the physics of mechanics and spin in nanotubes (and graphene) is important and opens possibilities for new quantum experiments.

The authors show data indicating the presence of Pauli-spin blockade and the formation of a spin-valley qubit. By driving the system at RF frequencies through the gates, they see features in the transport current at the mechanical frequency of the carbon nanotube and compare these features with a theoretical model where the non-linear mechanical classical motion is coupled to the qubit. The results are interesting and the theoretical model matches well the data they present.

The manuscript is well written overall. This work may perhaps be published in a high-impact journal such a Nature Communications provided that the authors respond the following questions in a convincing way.

Reply: We are sincerely grateful to the reviewer for their positive and encouraging evaluation of our work. We value their constructive suggestions and will address them carefully to strengthen the manuscript.

1 – While the data in Fig.2B can be related to spin-mechanics coupling, is it possible to explain the data by charge-mechanics coupling without any direct spin-mechanics coupling? The mechanical vibrations are observed away from the EDSR resonance, indicating charge-mechanics coupling. The combination of the electrical current shift due to mechanical vibrations (away from the EDSR resonance frequency) and of the current related to the EDSR resonance (away from the mechanical frequency) could lead to the impression of some sort of coupling even if there is not spin-mechanics coupling.

Reply: We thank the reviewer for their insightful comment, while we agree that in general both spin-mechanical and charge-mechanical coupling are present in our system, we are confident that these two effects lead to different signatures in the EDSR current.

A clear example is the one given in Fig. 3A of the main manuscript that exhibits a signature of spin-mechanical coupling as a bending of the EDSR resonance whilst approaching the mechanical resonance of the carbon nanotube (CNT) resonator. To highlight the difference with a typical charge-mechanical signature, we have now included a new section in the Supplementary information (S4) which shows an additional measurement, taken in the vicinity of the 2nd mechanical mode (see Supplementary Fig. 4). In this plot while we can clearly see the effect of charge-mechanical coupling with the EDSR current peak, as a softening and hardening of the mechanical resonance, the bending of the EDSR resonance (which we identified as a signature of spin-mechanical coupling) is absent.

A second point is that our spin mechanical model allows us to extract parameters from Fig. 3A of the main text (the resonant spin mechanical case), which allows us to reproduce qualitatively, with excellent agreement, the data of Fig. 2B. Therefore, the model we present is the minimal model we have found that satisfactorily explains all the key features observed both in the resonant and non resonant case.

Changes: We have added a new Section in the Supplementary Information, now Section S4, presenting more data about the presence of charge-mechanical coupling in our system and commenting on the differences with the observed spin-mechanical coupling.

2 – Is Fig.3a which is associated by the authors to a resonant coupling between the mechanical vibrations and the spin (as claimed in the Fig3 title), are the data really indicating resonant coupling in the sense that e.g. the quantum is oscillating between the mechanics and the spin? Or are the measurements related to the interplay between the drive of the mechanics and the drive of the spin without any direct spin-mechanics coupling? These are fundamentally two different phenomena.

Reply: There are no coherent oscillations of excitations between the mechanics and the qubit since such a phenomenon, typical of a Jaynes-Cummings-like Hamiltonian with transverse spin-oscillator coupling, is fundamentally different from the kind of coupling that we observe here, which is instead *longitudinal* (see Eq. 2 of the main text). This coupling has the effect of producing modulations of the Larmor frequency of the two-level system due to the mechanical displacement, which is responsible for the observed frequency shift in Fig. 3A.

In the case of “non-resonant” coupling, the frequency of the mechanical oscillator (262 MHz) is drastically different from the frequency of the two-level transition, which is 5.1 GHz (see Fig. 2A- B). We test this case by applying two different drives, one resonant with the CNT mode at 262 MHz and the other, at 5.1 GHz, resonant with the two-level system and far away from any other observed resonances of the mechanics.

What we denote as “resonant” coupling is the situation in which the frequency of the mechanical motion is exactly the same as the frequency of the two-level transition that is being driven. With a mechanical frequency of 262 MHz and $g \sim 0.3$ this occurs at a magnetic field $B \sim 60$ mT. In this case, a single continuous wave drive is applied on gate G2, at the unique resonant frequency of 262 MHz.

In such a resonant case, that of Fig.3A in the main text, there is thus only a single drive in the system, making us confident that the measurements are not simply the product of interplay between two drives without spin-mechanics. This can also be seen from our reply to question 1, where we show that for a different mechanical mode, the EDSR resonance shift of Fig. 3 is absent.

Changes: We have clarified the notation about the use of a single drive tone for the resonant case (see line 155 of the main text) and added a new sentence specifying the longitudinal nature of the spin-mechanical coupling and its difference with a conventional Jaynes-Cummings model (see lines 167-170).

3 – The coupling λ_1 is given with high precision 433 neV/nm. What is the error bar? How is calibrated the mechanical displacement?

Reply: The quoted value is a parameter selected to match the EDSR broadening observed in the simulation in Fig. 2C, and therefore is reported without an uncertainty (all other parameters are given in Eq. (17) of Section S7 of the Supplementary Information).

As we now more clearly specify in the main text, this parameter can also be estimated from the parameters estimated from the fit of the EDSR resonances in Fig. 3A of the main text, yielding a range of 380-455 $\frac{\text{neV}}{\text{nm}}$. This range can be regarded as the uncertainty for λ_1 (lines 185-186), and is indeed consistent with the value used in the simulation of Fig. 2B of the main text.

Regarding this estimate for λ_1 , the referee is right that its error is indeed dominated by the uncertainty in the estimation of the mechanical displacement. From the measurement of the frequency shift in Fig. 3A we obtain a fit of the total frequency shift, which is equal to $\Delta f = \lambda_2 \chi^2$ (see Eq. 2). In this way, we obtain the value of the product of coupling and displacement for the two resonances equal to $\Delta f^{(1)} = 40.3 \pm 0.8$ MHz and $\Delta f^{(2)} = 23.1 \pm 0.9$ MHz. To then obtain the bare coupling λ_2 requires an estimation of the amplitude of the CNT displacement χ . This is a significant source of uncertainty since a precise calibration of the displacement is not possible in this setup. What we have done is to use the best estimate we previously obtained for the displacement [1], which is in the order of $\chi = 1.5$ nm and is in line with other estimates in the literature for similar devices [3, 4]. Thus, as specified in the Supplementary Information, for an estimated displacement of 1.5 nm we obtain the bare quadratic couplings of approximately $\lambda_2^{(1)} = 76 \frac{\text{neV}}{\text{nm}^2}$, and $\lambda_2^{(2)} = 54 \frac{\text{neV}}{\text{nm}^2}$. As now further clarified at line 179 of the main text, from these values we can estimate the coefficient g_{\perp} and the resulting range of values for λ_1 .

Changes: We have adjusted the caption of Fig. 2 to clarify the arbitrary choice of the λ_1 parameters and further clarified this in the main text at lines 133-136. We have also modified lines 179, 185-186, to clarify the major contribution of the displacement error in defining the uncertainty of the estimated couplings $\lambda_2^{(1)}$ and $\lambda_2^{(2)}$. We have also changed Eq. (18) in Section S7 of the Supplementary Information to include the estimated uncertainties.

4 – Why is it necessary that the nanotube is brought to the non-linear regime (Duffing) to see a signature of the spin-mechanics coupling?

Reply: The non-linear mechanical regime is not a requirement to have spin-mechanics coupling. Indeed, from our understanding of the origin of the coupling and the model presented in the article, the coupling

could in principle also be seen in the linear regime. With that said, it is well known that CNTs are easily driven into the non-linear regime [5], and given that our estimated bare coupling is quite small, relatively strong driving of the CNT was necessary to produce a strong enough signature in the EDSR signal. This is why all of our measurements are in the non-linear mechanical regime.

Changes: To avoid confusion, we have further clarified this aspect in the main text by adding a sentence at lines 129-130.

These questions that are less critical should also be addressed.

5 – *What are the free parameters in the theoretical model to fit the power-dependent data in Fig. 2?*

Reply: We respectfully remark that we do not fit the data in Fig. 2 nor of other power-dependent data in the main text and supplementary. The parameters we used to obtain the numerical simulation that we present in Fig. 2C of the main text, which we report below, are also given in Eq. (17), section S7 of the Supplementary Information. These are:

$$\lambda_1^{(2)} = 433 \frac{\text{neV}}{\text{nm}}, \quad \bar{\beta} = 2 \times 10^{-12}, \quad \Gamma = 392.9 \text{ MHz}, \quad \Omega = 2\pi \times 7.6 \text{ MHz}, \quad (1)$$

where $\lambda_1^{(2)}$ is the first order spin mechanical coupling, $\bar{\beta}$ is the non linearity coefficient, $\Gamma \sim 1/T_2^*$ is the decoherence rate of the spin and Ω is the Rabi frequency. As detailed in Section S7 of the Supplementary Information, the free parameters $\lambda_1^{(2)}$ and $\bar{\beta}$, were chosen to reproduce respectively the broadening and asymmetry of the EDSR peak in Fig. 2B (main text) near the mechanical resonance, while Γ and Ω were set to match the EDSR peak width away from the resonance.

6 – *The text reads “the charge transition $(N, 1)-(N + 1, 0)$ ” in a p-n configuration. We understand that $(N, 1)$ corresponds to N holes in the left dot and 1 electron in the right dot. Should the text not read $(N, 1)-(N - 1, 0)$?*

Reply: We thank the reviewer for spotting this typo; indeed, the charge configuration should have read $(N, 1)-(N-1, 0)$.

Changes: We have made appropriate corrections to the main text by also taking into account the reviewer’s concerns regarding a later question (please see our reply to the reviewer’s question 10), as well as concerns from reviewers 2 and 3.

7 – *In Fig 1F, how do you define experimentally zero detuning?*

Reply: We identify the 0-detuning line with the baseline of the bias triangle extracted from measurements such as that of Fig. 1C of the main text.

Changes: We have addressed this question (together with question 12 from the reviewer) by slightly modifying Fig. 1 of the main text (highlighting better the baseline of the bias triangle) and explaining in the caption the definition of 0-detuning point.

8 – *(line 191): How can the nanotube length and radius be estimated from previous experiments?*

Reply: In one of our previous works [1], which was done on the same device and during the same cooldown, we estimated these geometric parameters by fitting the dispersion of the CNT mechanical frequency as a function of the electrostatic tensioning via gate voltages (see Fig. 7 and Table II of Ref. [1]). To extract the effective mass and geometric properties of the CNT, we fit the dependence of the mechanical resonance on gate voltage as derived in Ref. [2].

Changes: We have added a brief description in the Methods to clarify this aspect.

9 – *(line 29) At least one reference should be provided*

Reply: We thank the reviewer for noticing this, we have added a reference as requested.

Changes: We added the requested reference (Ref.[6]) at line 36 of the main text.

10 – (line 58) Looking at the charge stability diagram of Supplementary Fig.1, it is not clear that the transition is “ $(N-1,1) \rightarrow (N,0)$ ”

Reply: Our statement about the charge states comes from our reading of Fig. 1 of the Supplementary, where we interpret that there is a bandgap around 0V. However, we appreciate that this might not be a fully conclusive measurement. As such, and given that none of the modelling and analysis done in the paper depends on which charge transition we are operating at (so long as PSB is observed), we have removed unnecessary claims on the nature of the charge carriers in our systems and adapted the main text accordingly, see lines 68-70 of the main text. We have made consistent changes also in the relevant Supplementary Information section and the caption of the Supplementary Fig. 2.

Changes: See our previous reply to question 6 for the relevant changes.

11 – (Line 61) What is the orientation of the magnetic field B_X ? Is it along the nanotube axis? Has this been calibrated? Have the authors tried some other orientations? There is some extra information in the supplemental material but this should be clearer in the main text.

Reply: B_X is oriented at approximately 55° from the nanotube main axis (see Fig.1 of the main text and new Supplementary Fig. 2), whose direction is identified by B_{\parallel} in the main text. This angle was calibrated via magneto spectroscopy measurements, see Supplementary Fig. 2A of the Supplementary Information. The measurement shows that PSB is lifted when the magnetic field is oriented at approximately 55° between the X and Z directions of the laboratory frame. As shown in Ref. [7], this is evidence that the direction of the field is parallel to the nanotube axis. Moreover, this value is consistent with the expected range of angles based on the nanotube orientation within our setup, accounting for uncertainties due to imperfect alignment of the CNT across the trench.

Changes: To clarify this aspect, we have modified Fig. 1A of the main text, and Fig. 2S of the Supplementary Information. In the caption of Fig. 1A and at line 148 of the main text, we have also added a reference to the relevant section of the Supplementary, where the field orientation and its calibration are discussed in detail.

12 – (Line 62) The suppression of the triangle baseline is not at all clearly seen in Fig. 1B/C.

Reply: We thank the reviewer for raising this point.

Changes: We have added a white dashed line at the baseline of the bias triangle to make this more visible and adjusted the figure caption accordingly.

13 – (Line 66) In order to state a spin-valley qubit is being defined, either spectroscopy or time-resolved measurements of the qubit should be provided, also giving information about its corresponding decoherence and possible multilevel character.

Reply: Previous work in the literature, both theoretical and experimental [8, 9, 10], shows that for typical devices within the range of magnetic field values explored here, the spin-orbit coupling is a significant energy scale and spin-valley hybridisation must be accounted for. Indeed, only at large magnetic field values, e.g. such as 1.2 T as shown in Ref. [11], can one go into the regime of a pure spin-qubit where valley can be ignored.

Changes: We have clarified our claim at line 75-76 of the main text.

14 – (Line 66) I recommend adding some more details clarifying the time-resolved protocol.

Reply: We thank the reviewer for raising this issue.

Changes: We have modified the main text at lines 76-83 to include further details about the time-resolved protocol.

15 – (Line 80) *In Fig.1 1F, is the feature observed at 262 MHz the only mechanical resonance in the entire frequency spectrum?*

Reply: At higher frequency sweeps, we also observe well-resolved resonances corresponding to higher mechanical modes, with harmonics at 512 MHz, 1.022 GHz, 1.530 GHz, and 2.07 GHz. However, because these modes are widely separated in frequency, have high-quality factors, and the measurement drive tones have small bandwidth, we restrict our analysis to a single mechanical mode.

Changes: With reference to our reply to the reviewer’s first question, we have now added a plot in Section S4 of the Supplementary Information, showing data exhibiting signatures of charge-mechanical coupling with a higher mode around 512-514 MHz.

16 – (Line 116) *The authors should stay at some point that the mechanical non-linearity is probably due to the high driving power and not to the coupling of the spin.*

Reply: We agree with the referee that indeed the non-linearity is due to the strong driving and not the coupling with the spin.

Changes: We have added a sentence to clarify this at line 108 of the main text.

Reviewer 2

Reply: We are very grateful to the reviewer for their evaluation of our work.

Reviewer 3

In this manuscript the authors demonstrate the coupling of a single spin with the bending motion of a carbon nanotube. Several spin transitions are identified via EDSR. The coupling between such spin transitions and the mechanical degree of freedom of the nanotube is observed in two distinct regimes. In the off-resonant regime, the considered spin transition is a few GHz away from the mechanical bending mode, and the main observed effect is a mechanically induced broadening of the EDSR resonance. In the resonant regime, two neighbour spin transitions could be brought into resonance with the mechanical mode and both of them undergo a frequency shift of their associated EDSR peak. The authors also provide a model describing the spin-mechanical coupling relying on the anisotropy of the g-factor. This model could explain very well both observed behaviours; namely, an increased dissipation in the non-resonant regime and a frequency shift in the resonant regime.

I think those results are remarkable and represent a long-standing goal in the community. I strongly support the publication of this work in a high impact journal such as Nature Communication provided the authors answer the following questions:

Reply: We are sincerely grateful to the reviewers for their positive evaluation and for highlighting the importance of our work. Below we have addressed their feedback and suggestions and hope that they will find them to enhance the quality of the manuscript.

The mechanical origin of the signal at around 262MHz is attested by the bifurcation behaviour which is characteristic of bending mode in carbon nanotubes. Do you have extra data about the mechanical resonance, such as power dependence or a broad range frequency scan? Such extra measurements could be useful to know whether it is the fundamental mode or not.

Reply: In Ref. [1], which uses the same device in the same cooldown, we report a characterisation of the dependence of the frequency of the fundamental mode as a function of gate voltage. The data is used to fit the characteristic Euler-Bernoulli equation as described in Ref. [2] to extract the mass and geometric parameters of our CNT device, (see also our reply to question 8 of reviewer 1). Moreover, in Supplementary Fig. 3F and 3G of the Supplementary Information, we show the power dependence of the fundamental mechanical mode, showing a broadening of the resonance due to Duffing as well as a shift to higher frequencies. Finally, we have done larger frequency sweeps and we have seen higher modes of the mechanics (for example, see new Fig. 4 of the Supplementary information, and our reply to question 15 of reviewer 1). The resonance at 262 MHz is the lowest we could identify, suggesting it is likely to be the fundamental mode. It is important to note that none of the modelling and analysis done relies on any assumption about which mode we are coupling to.

Changes: As mentioned previously, we have now added a new Section (S4) in the Supplementary Information showing data highlighting charge-mechanical coupling with a second higher frequency mode. In the Methods section, we have also clarified where other characterisation measurements of our device can be found within existing literature.

Could you give an estimation of the spin-orbit coupling in the nanotube based on the g-factor anisotropy? Since this anisotropy is at the core of the coupling mechanism, it could be interesting to discuss its origin a bit more.

Reply: The referee raises an interesting point, which highlights the rich physics that can be explored in this system. While the spin-orbit coupling is at the heart of the g-factor anisotropy, this is not the only contribution to the level structure of spin-valley qubits. Indeed, the effective g-factors depend non-trivially on the spin-orbit coupling, the inter-valley coupling of each dot, and the orbital magnetic moment (see Refs. [8, 10]). Although estimating the spin-orbit energy scale in an ultraclean nanotube quantum dot is possible [12], our device does not show the characteristic signatures of ultraclean devices. To our knowledge, no established models exist for extracting the spin-orbit energy scale in devices with similar disorder or complexity to ours. For future work, we do agree with the reviewer that our system is indeed promising to further investigate the origin of the g-factor anisotropy in carbon nanotubes and systems with strong spin-orbit interaction in general.

Changes: We have added a sentence at line 195-196 of the main text, highlighting this in the main text.

From the charge stability diagram presented in the Supplementary Fig. 1, it is not clear to me where is the semiconducting gap along the gate V_{G5} . Estimating an absolute electron number seems a bit audacious here. A broader map could maybe help to identify with more reliability the p-n and p-p quadrant. Do you have such a 2D-map?

Reply: We appreciate that the data in the Supplementary Fig. 1 might not be a fully conclusive measurement for the presence of the bandgap. As such, and given that none of the modelling and analysis done in the paper depends on which charge transition we are operating at (so long as PSB is observed), we have modified the main text and Section S1 of the Supplementary Information accordingly.

Changes: We have adjusted the main text and relevant section in the supplementary section to avoid any claim regarding the nature of the charge carriers in our device. Please see also our reply to questions 6 and 10 of reviewer 1.

In figures 2A, 2B, and 3A the frequency is swept from high to low. Why? I would expect a stronger signal if it were swept in the opposite direction due to the Duffing behavior of the bending mode of the nanotube.

Reply: We have decided to include a consistent choice of sweep direction throughout the manuscript. Both for the resonant and non-resonant cases discussed, we have some data in both sweep directions, which show no appreciable difference. The results are qualitatively the same, and the only appreciable change is in the frequency at which the jump in the mechanical response occurs, consistent with the hysteresis behaviour in Duffing oscillators.

Changes: We have now included a short section (Section S5) in the Supplementary Information showing data of the off-resonant and resonant signatures of spin mechanical coupling with opposite sweep directions.

I found the naming of some frequencies a bit confusing, such as f_{osci} and f_{spin} which are two driving frequencies. A subtext “d,...” or “drive,...” would clarify their meaning. Same for P_{osci} .

Reply: We agree with the referee and we implemented the requested change throughout the manuscript.

References

- [1] F. Vigneau, J. Monsel, J. Tabanera, K. Aggarwal, L. Bresque, F. Fedele, F. Cerisola, G.A.D. Briggs, J. Anders, J.M.R. Parrondo, A. Auffèves, and N. Ares, Ultrastrong coupling between electron tunneling and mechanical motion, *Phys. Rev. Res.* **4**, 043168 (2022). <https://doi.org/10.1103/PhysRevResearch.4.043168>
- [2] S. Sapmaz, Ya. M. Blanter, L. Gurevich, and H. S. J. van der Zant, Carbon nanotubes as nanoelectromechanical systems, *Physical Review B* **67**, 235414 (2003).
- [3] C. Urgell, W. Yang, S.L. De Bonis, C. Samanta, M.J. Esplandiu, Q. Dong, Y. Jin, A. Bachtold, Cooling and self-oscillation in a nanotube electromechanical resonator. *Nature Physics* **16**, 32-37 (2020).
- [4] C. Samanta, S. L. De Bonis, C. B. Møller, R. Tormo-Queralt, W. Yang, C. Urgell, B. Stamenic, B. Thibault, Y. Jin, D. A. Czaplewski, F. Pistolesi and A. Bachtold, *Nature Physics* **19**, 1340–1344 (2023)
- [5] A. Castellanos-Gomez, H.B. Meerwaldt, W.J. Venstra, H.S. J. van der Zant, and A.G. Steele, Strong and tunable mode coupling in carbon nanotube resonators, *Physical Review B* **86** 041402 (2012).
- [6] O. Arcizet, V. Jacques, A. Siria, P. Poncharal, P. Vincent, and S. Seidelin, A single nitrogen-vacancy defect coupled to a nanomechanical oscillator, *Nature Physics* **7**, 879 (2011).
- [7] J. Danon, Y.V. Nazarov, Pauli spin blockade in the presence of strong spin-orbit coupling, *Phys. Rev. B* **80**, 041301 (2009).
- [8] A. Palyi, P.R. Struck, M. Rudner, K. Flensberg, and G. Burkard, Spin-Orbit-Induced Strong Coupling of a Single Spin to a Nanomechanical Resonator, *Phys. Rev. Lett.* **108**, 206811 (2012).

- [9] T. Pei, A. Palyi, M. Mergenthaler, N. Ares, A. Mavalankar, J. H. Warner, G.A.D. Briggs, and E.A. Laird, Hyperfine and Spin-Orbit Coupling Effects on Decay of Spin-Valley States in a Carbon Nanotube, *Phys. Rev. Lett.* **118**, 177701 (2017).
- [10] E.A. Laird, F. Kuemmeth, G.A. Steele, K. Grove-Rasmussen, J. Nygard, K. Flensberg, and L.P. Kouwenhoven, Quantum transport in carbon nanotubes, *Rev. Mod. Phys.* **87**, 703 (2015).
- [11] H. O. H. Churchill, F. Kuemmeth, J. W. Harlow, A. J. Bestwick, E. I. Rashba, K. Flensberg, C. H. Stwertka, T. Taychatanapat, S. K. Watson, and C. M. Marcus, Relaxation and Dephasing in a Two-Electron ^{13}C Nanotube Double Quantum Dot *Phys. Rev. Lett.* **102**, 166802 (2009).
- [12] F. Kuemmeth, S. Ilani, D. C. Ralph, and P. L. McEuen, Coupling of spin and orbital motion of electrons in carbon nanotubes, *Nature* **452**, 448–452 (2008).

Manuscript reference: NCOMMS-25-08864-T
Coupling a single spin to the motion of a carbon nanotube

,

We sincerely thank all three reviewers for their positive evaluation of our work. As requested, we provide a detailed, point-by-point response below.

Yours sincerely,

The authors

Reviewer 1

The authors investigate spin–mechanics coupling in suspended carbon nanotubes. Owing to their low mass and large zero-point motion, nanotube resonators enable strong coupling to other degrees of freedom, while their high mechanical frequencies allow the possibility of ground-state operation at dilution temperatures. Nanotube spin physics is also rich and well established. This work presents the first experimental evidence of spin–vibration coupling. The authors observe Pauli-spin blockade and spin–valley qubit formation. The authors present two sets of data, one called off-resonant coupling and the other called resonant coupling, and these two data sets can be described by a model using the same parameters. The manuscript is clearly written and I recommend publication. However, the outlook in the abstract and conclusion seems somewhat overstated in light of the present results.

Reply: We thank the Reviewer for their positive assessment. As suggested, we have revised the statements in the abstract and conclusion. These changes are highlighted in the edited pdf of the revised manuscript.

Reviewer 2

Reply: We are sincerely grateful to the Reviewer for their positive evaluation of our work.

Reviewer 3

I am satisfied with the reply of the authors and fully recommend the publication in Nature Communications.

Reply: We thank the Reviewer for their positive feedback on our work.